# SKG-Lock+: A Provably Secure Logic Locking Scheme Creating Significant Output Corruption

**Quang-Linh Nguyen** [†]**, Sophie Dupuis** *[ID]**, Marie-Lise Flottes** [ID] **and Bruno Rouzeyre**

Laboratoire d'Informatique, de Robotique et de Microélectronique de Montpellier, Université de Montpellier, CNRS, CEDEX 5, 34095 Montpellier, France

* Correspondence: sophie.dupuis@lirmm.fr; Tel.: +33-499-585-134
† Current affiliation: STMicroelectronics, 12 Rue Jules Horowitz, 38019 Grenoble, France.

**Abstract:** The current trend to globalize the supply chain in the Integrated Circuits (ICs) industry has raised several security concerns including, among others, IC overproduction. Over the past years, logic locking has grown into a prominent countermeasure to tackle this threat in particular. Logic locking consists of "locking" an IC with an added primary input, the so-called key, which, unless fed with the correct secret value, renders the ICs unusable. One of the first criteria ensuring the quality of a logic locking technique was the output corruption, i.e., the corruption at the outputs of a locked circuit, for any wrong key value. However, since the introduction of SAT-based attacks, resulting countermeasures have compromised this criterion in favor of a better resilience against such attacks. In this work, we propose SKG-Lock+, a Provably Secure Logic Locking scheme that can thwart SAT-based attacks while maintaining significant output corruption. We perform a comprehensive security analysis of SKG-Lock+ and show its resilience against SAT-based attacks, as well as various other state-of-the-art attacks. Compared with related works, SKG-Lock+ provides higher output corruption and incurs acceptable overhead.

**Keywords:** logic locking; SAT attack; Design-for-Trust; hardware security; IP protection; overproduction

## 1. Introduction

In today's semiconductor industry, outsourcing has become the prevailing business model [1]. Outsourcing and offshoring the fabrication process in particular has been a major trend for decades, due to ever-increasing manufacturing costs, leading ultimately to an increased exposure of Intellectual Property (IP) and Integrated Circuits (ICs) to external—possibly unreliable—actors. Due to this loss of control, combined with an increasing risk due to ever-growing adversarial capabilities throughout the supply chain, several threats, such as IC overproduction, counterfeiting, IP piracy and Hardware Trojan insertion, have become major sources of concern [2].

Numerous Design-for-Trust approaches have introduced preventive mechanisms over the last years [3]. One of the most prominent approaches against IC overproduction and IP piracy is logic locking [4,5]. Logic locking is based on adding key inputs into a design so that the circuit behaves as expected only in the presence of the correct secret key value. For incorrect keys, circuit's outputs are corrupted and provide erroneous data. The correct key is programmed post-fabrication by a trusted party thereby remaining unknown to manufacturers and end-users. Therefore, logic locking helps design houses protect their IPs from untrusted entities. The first logic locking technique was proposed in 2008 [6], introducing the concept of XOR/XNOR key-gates insertion at random locations inside a design (RLL) (cf. Figure 1). The following works proposed different types of key-gates (AND/OR gates, MUXes, LUTs, etc.) and insertion strategies [7–11]. The most important insertion criterion, introduced by Fault-based Logic Locking (FLL) [7], aims to provide sufficient errors observed at a locked circuit' outputs, referred to as high output corruption.

However, such key-gate-based logic locking techniques have been reported to be highly susceptible to the so-called *SAT attack* in 2015 [12], which represents a milestone in the logic locking research area. This attack uses a Boolean satisfiability (SAT) solver and an unlocked circuit, referred to as the *oracle*, to iteratively and efficiently prune out wrong key values. Most subsequent logic locking techniques aimed at counteracting the SAT attack, by exponentially increasing its computing time [13,14]. In other words, thanks to the use of a one-point function, the SAT attack could only eliminate one key value per iteration, which made it as ineffective as a brute-force attack. However, to do so, output corruption was dramatically diminished [15], making the ICs mostly functional despite being supplied with a wrong key value. Furthermore, these Provably Secure Logic Locking (PSLL) schemes introduced an isolated block leaving structural traces allowing the identification of the block by an attacker [16]. How to improve these PSLL schemes has been an extremely prolific field of research over the last years [15,17–23], aiming at reaching an acceptable trade-off between attack (both functional/SAT-based and structural-based) resilience, output corruption, and area overhead. This research has also been coupled with improvements in attacks, mimicking an extremely prolific cat and mouse game, which motivates all the more the proposal of new protection methods, while making this task even more challenging.

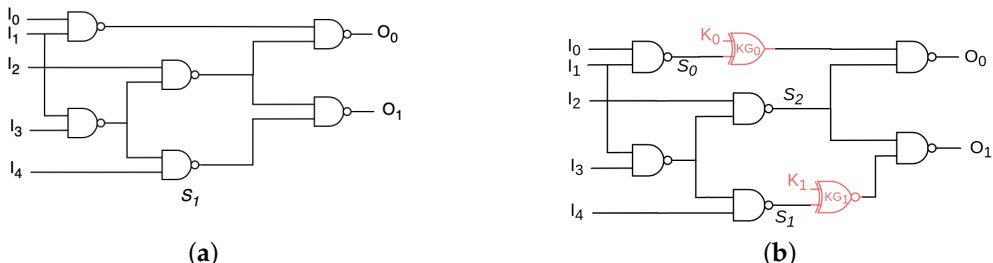

**Figure 1.** Logic locking technique: (**a**) original circuit, (**b**) key-gate insertion based logic locking.

To address the limitations of previous schemes, we proposed SKG-Lock [24], featuring the characteristics of causing corruption at multiple signals with different corruptibilities. In this paper, we present SKG-Lock+, which also provides high security against attacks while further improving corruption. The two main contributions of this work are as follows:

1.  We present an improved structure of the so-called switch controller with respect to the one introduced in SKG-Lock, which further improves output corruption;
2.  We propose a new key-gate insertion strategy $F_P LL$, which, based on the computation of signal probabilities, allows maximizing output corruption, with a far shorter computation time than previously proposed methods.

Furthermore, we provide a thorough security analysis of SKG-Lock+ against oracle-guided attacks, proving that SKG-Lock+ achieves maximum resilience against the SAT attack regardless of different corruptibilities of the switchable key-gates, as well as its evaluation in terms of attack resilience, output corruption and overheads, along with comparisons with state-of-the-art techniques.

The rest of the paper is organized as follows. Section 2 provides background on criteria for logic locking, SAT attack and its countermeasures, and recent works on both attacks and defenses. Section 3 presents our proposed logic locking scheme, SKG-Lock+. Security analysis of SKG-Lock+ against state-of-the-art attacks is discussed in Section 4. Experimental results are shown in Section 5, in terms of computation time, output corruption, resilience against publicly available attacks and area/delay overheads. Discussions and conclusion follow in Sections 6 and 7.

## 2. Preliminaries

### 2.1. Output Corruption

Output corruption means that a locked circuit behaves (sufficiently) erroneously upon application of any wrong key value. In order to be properly quantified, it should be further categorized into several criteria as follows.

Output corruptibility—often termed Hamming distance [7] due to how it is computed—is estimated by the average Hamming Distance *HD* between the outputs of a locked and an unlocked circuit. For our experimentation, we applied to a locked circuit $N_K$ key values, each with $N_I$ input patterns, and observed its *m*-bit output $O_L$ along with the output $O$ of the original circuit with the same input patterns, to compute the *HD* as follows:

$$\frac{1}{N_K \times N_I \times m} \sum_{i=1}^{N_K} \sum_{j=1}^{N_I} HD(O_L(I_j, K_i), O(I_j)) \times 100\% \tag{1}$$

Output corruptibility is the original and most often used criterion to quantify output corruption. Its targeted value is 50% for maximum obscureness for an attacker. However, even a good output corruptibility may not be sufficient for preventing the usage of locked circuits. Among others, a good output corruptibility does not indeed ensure that all outputs can be impacted. In the image processing domain, for instance, a locked circuit could still be used, e.g., if the more significant bits of the output are not impacted.

Output corruption rate—also termed output error rate or error rate [17,25]—presents the probability of observing erroneous bit(s) at the output vector of a locked circuit. It is measured by the percentage of input patterns that lead to errors at circuit outputs when any incorrect key value is applied:

$$\frac{1}{N_K \times N_I} \sum_{i=1}^{N_K} \sum_{j=1}^{N_I} c \times 100\% \tag{2}$$

where,

$$c = \begin{cases} 1, & \text{if } HD(O_L(I_j, K_i), O(I_j)) \geq 1 \\ 0, & \text{otherwise} \end{cases} \tag{3}$$

Output corruption coverage presents the magnitude of corruption propagated to circuit outputs. It is characterized by the maximum number of output bits that can be corrupted, i.e., the maximum Hamming distance between the outputs on applying any wrong key and the correct key:

$$\frac{l}{m} \times 100\% \tag{4}$$

where,

$$l = max(HD(O_L(I_j, K_i), O(I_j))) \forall i \in [1...N_K], j \in [1...N_I] \tag{5}$$

To complement output corruptibility in a relevant way, output corruption rate (100% target) ensures that all input patterns lead to corruption and output corruption coverage (100% target) ensures that there is no output that is never corrupted.

### 2.2. Threat Model

Logic locking aims at preventing an untrustworthy foundry from selling overproduced ICs on the black market, by rendering those inoperable, until they are properly activated. The *attacker* is therefore the foundry—or at least a rogue employee in the foundry—that, as a consequence, aims at unlocking the ICs/circumventing the protection (for example by finding the value of the secret key). Two types of attacks have been proposed.

*Oracle-guided attacks*—such as the SAT attack—whose threat model is described in Figure 2, assume that the attacker has access to two fundamental assets:

- The locked netlist, i.e., the netlist containing the logic locking structure—possibly obtained from reverse-engineering (e.g., the manufacturer can obtain it from the layout of the logic locked design);
- An oracle, i.e., an unlocked IC (with accessible scan chains).

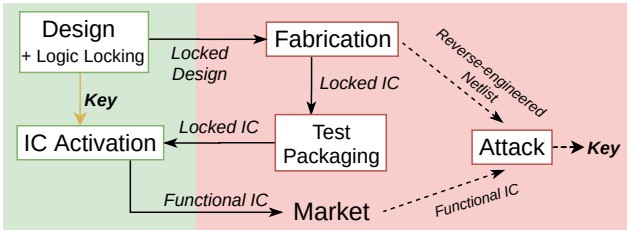

**Figure 2.** IC supply chain and threat model on logic locking.

Oracle-less attacks, on the other hand, assume that the attacker does not have access to an oracle, only to the locked netlist. Structural attacks belong to this category, including removal attacks aiming to detect the protection logic to remove/disconnect it.

Note that, in this paper, we also assume that the physical security of the key—e.g., stored in a tamper-proof memory—to prevent direct access by the attacker is guaranteed. The veracity of this assumption, which may be undermined by attacks such as optical probing [26], is out of the scope of this paper. Furthermore, one should notice that logic locking methods generate a unique global key value. Logic locking security can nevertheless be improved thanks to the individualization of the key value for each IC, by using the pre-step scheme, using process variations, provided by PUFs for example [7].

### 2.3. The SAT Attack and "Post SAT" Related Works

#### 2.3.1. The SAT Attack

The SAT attack [12] is an iterative process, as presented in Figure 3. In each iteration, the attack finds a so-called Distinguish Input Pattern (DIP), i.e., an input pattern that results in different output values for two different key values. The DIP is then applied to the oracle to prune out the wrong key values. Note that the attack is able to prune out all key values that generate corruption for this particular input pattern—referred to as equivalent keys or keys of an equivalence class. By adding the observed disagreement between the locked netlist and the oracle as new constraints to the SAT solver, the attack reduces the key search space iteratively until no more DIP can be identified. Then, the SAT solver deduces the correct key.

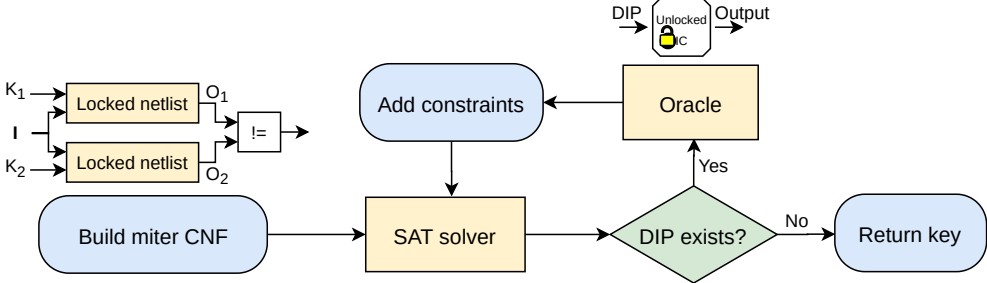

**Figure 3.** The SAT attack [12].

This attack represents a milestone in logic locking research. Most of the methods released since 2015 attempt to be effective against this attack, alongside new attacks. Three tracks were studied to prevent the attack:

1. Prevent it from being launched, e.g., by preventing the formalization of the circuit into a directed acyclic graph. This can be done with the introduction of cyclic interconnection [27] (countered by an enhanced SAT version assuming that "there is one correct key that will generate an acyclic circuit" [28]). The idea of delay locking has also

been introduced in [29], in which the key determines the correct functionality along with the correct timing profile of a circuit (also countered by the SMT enhanced SAT attack [30]). In the particular case of sequential circuits, which need to be modeled as combinational ones thanks to the use of scan-chains, countermeasures based on the protection of scan chains exist also, thereby preventing the use of the oracle [31–33]. However, this kind of solution may be suitable to all defenders, e.g., those who do not have control over the test infrastructure of their designs.

2.  Lengthen tremendously the attack computational time:

    (a)  By increasing the computational time of each iteration. To that end, researchers have proposed countermeasures based on SAT-hard structures, such as cryptographic ciphers [10,34], nevertheless having impractically high overheads. Full-Lock [35] introduces another kind of SAT-hard structure, programmable logic and routing blocks (countered by an SAT-based attack with advanced modeling techniques [36]). However, an SAT-based attack guided by a neural-network has recently been proposed to solve such structures, as multipliers, crossbars, LUTs and AND-trees [37].

    (b)  By increasing the number of iterations needed by the attack. It is this research direction that has been the most studied, and of which our proposal is a part. We therefore detail it in the following sub-sections.

### 2.3.2. Provably Secure Logic Locking

The first provable SAT countermeasures were SARLock [13] and Anti-SAT [14]. As already introduced, they use a point-function (i.e., a boolean function that outputs the value '1' for exactly one input pattern) based structure inserted beside the circuit (cf. Figure 4a) to maximize the number of iterations of the SAT attack to $2^n$ ($n$ being the key size): the point-function only corrupts the circuit output for one corresponding key value per input pattern [17]. Let us define the SAT resilience level of a logic locking technique, which is $n$-secure if the number of iterations returned by the SAT attack on its locked circuits is $2^n$.

Since output corruption is dramatically low with this type of technique, a naive way to increase corruption is to insert additional key-gates beside the point-function lock [13,14]. In these so-called *compound schemes*, the key is divided into two distinct parts (cf. Figure 4b): the strength of the solution against the SAT attack is provided by the key bits dedicated to the SAT protection, whereas the other key bits are dedicated to the increase of the output corruption.

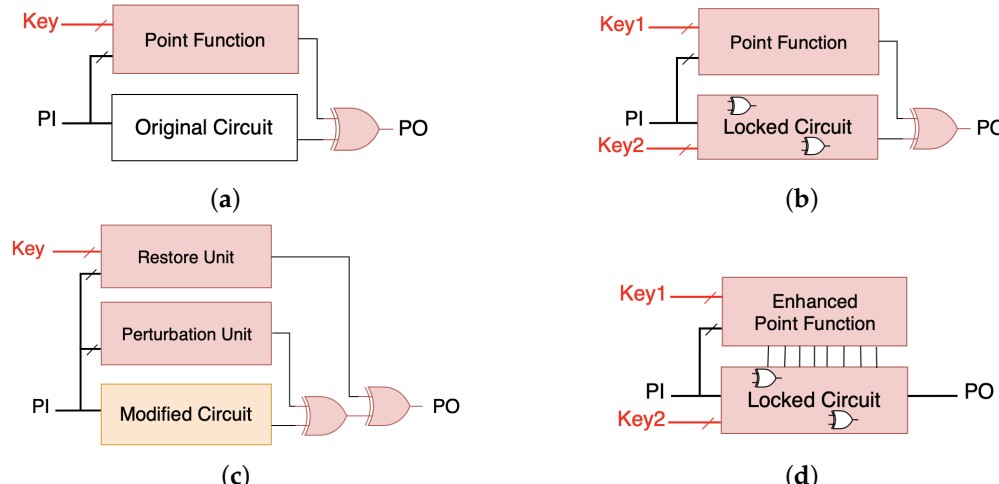

**Figure 4.** General structures of PSLL: (**a**) point-function based, (**b**) compound (point-function + logic locking), (**c**) CAC, (**d**) SKG-Lock+.

### 2.3.3. Post-SAT Attacks

Approximate attacks, notably AppSAT [15] and Double-DIP [38], are oracle-guided attacks that aim at finding an approximately correct key, i.e., a key value for which output corruption of the locked circuit is very low. Due to additional capabilities such as error estimation and random query reinforcement, they can avoid being trapped into solving the point-function. The Bypass attack [39] aims to build a bypass circuit to correct the output of a circuit locked by a point-function based technique. The attack collects disagreeing input patterns of two copies of the locked circuit supplied with different wrong key values. The bypass circuit is built to correct circuit outputs for these input patterns.

Regarding compound schemes, such attacks can be used, if combined with a pre-processing step to differentiate the key bits corresponding to the key-gate insertion technique to those corresponding to the SAT protection [40]. For example, both the SAT attack and the Bypass attack can be used on each part of the key. Fa-SAT has also been proposed [41], which takes advantage of the fault-injection principle to aid the SAT attack tackle compound schemes. Ref. [42] is a statistical-based attack that utilizes the variation in output corruption for different key bits to also tackle compound schemes.

Since point-function based blocks are connected to the circuit through a single signal, an attacker can aim at detecting the output of this block in order to remove it and retrieve the original design. Several oracle-less removal attacks [43,44] (initially dedicated to a particular protection) were proposed to exploit such vulnerabilities thanks to: (i) signal probability analysis (since the block output has exceptionally skewed probabilities), (ii) fanin analysis (since all key-inputs converge at this signal), (iii) partitioning algorithms (since the block is isolated and its size can be estimated).

### 2.3.4. Improved PSLL

In order to mitigate structural vulnerabilities of PSLL schemes or/and enhancing output corruption, new point-function structures have been proposed. Diversified Tree Logic (DTL) [15] provides tunable output corruption with the modification of a few gates in the point-function based block. By replacing some of the gates in an AND tree structure with OR/AND/XOR gates, the corruptibility of the added block can indeed be increased, generating an increase in output corruption and output corruption rate. Noise-based logic locking [21] introduces an improvement of the Anti-SAT block with non-complementary sub-blocks, which avoids a block output with probabilities skewed toward 0 or 1. By using non-complementary functions, a wide variety of structures can be implemented, less susceptible to removal attacks. CAS-Lock [22] proposes cascaded structures for the complementary sub-blocks. Instead of an AND tree structure, the cascaded structures in this technique contain AND and OR gates. The proposed CAS-Lock block increases exponentially the complexity for an SAT attack, while providing considerable output corruption. Its output corruption is tunable by changing the location and number of AND/OR gates in the sub-blocks. G-Anti-SAT [23] introduces a generalized approach to designing the SAT resilient logic lock. The work identifies a set of constraints for the function of each sub-block that can enable achieving maximum SAT resilience as well as non-trivial corruption. It then uses K-maps to implement such functions. A large variety of structures for sub-blocks can be realized, either complementary or non-complementary, AND tree or non-AND tree. Thus, Anti-SAT and CAS-Lock can be considered as special cases of G-Anti-SAT.

### 2.3.5. Corrupt-and-Correct Schemes

On the other hand, point-functions based on a Corrupt-And-Correct (CAC) scheme were also proposed, in order to mitigate structural vulnerabilities. TTLock [45] introduced the CAC concept, in which the original circuit is transformed so that its output is flipped for one so-called protected input pattern (which is also the correct key value): a perturbation unit consisting of a comparator is added to strip the functionality of the original design, and an additional restore unit is also inserted to correct the output in the presence of a correct key value (cf. Figure 4c). An attacker aiming at removing the added block

may be able to remove the restore unit, but not the perturb unit, thus obtaining a circuit still not functioning properly. Then, the Stripped Functionality Logic Locking (SFLL) method [17–19] generalised the concept, by e.g., allowing to choose the Hamming distance $h$ between the protected input patterns and the correct key (SFLL-HD) or to choose directly the $c$ protected input patterns (SFLL-flex). By increasing $h$ or $c$, the output corruption can grow, however, at the cost of decreasing SAT-attack resilience and increasing the area overhead.

### 2.3.6. Post-PSLL/CAC Attacks

Recently, oracle-guided and oracle-less attacks have been enhanced to tackle improved PSLL and/or CAC schemes. SFLL was attacked by the removal FALL attack [16], which manages to detect and remove its perturbation unit. CAS-lock was attacked by both an oracle-guided and an oracle-less attack in [46]. The SPI attack [47] is an oracle-guided and structural-based attack that exploits the properties of EDA tools to undermine CAC schemes. However, a property consisting of wisely choosing the protected input pattern(s) intrinsic to the CAC technique is also proposed by the authors to counteract the attack. The removal attack GNNUnlock+ [48] uses a graph neural network to identify all previously proposed PSLL methods. To do so, it identifies the specific and common characteristics of signals in the different blocks and, after training, the graph neural network is able to classify which signals belong to the protection blocks.

Other types of oracle-less attacks have also been introduced. Synthesis-based attacks aim to extract the secret key by synthesizing the locked netlist upon applying constraints on key-inputs. The SCOPE attack [49] tackles each key bit individually. For each key-input, it consists of making two circuit copies, each with logic 1 or 0 assigned to the key-input, then synthesizing and optimizing the two circuits, before comparing them using statistical analysis. The deduced key bit is the one associated with the more optimized circuit copy. Ref. [50] prunes out incorrect keys that introduce a significant level of logic redundancy. CLIC-A [51] is an ATPG-based attack, which exploits the use of a protected input patterns in CAC schemes.

## 3. Proposed Logic Locking Scheme: SKG-Lock+

In line with proposals based on improved point-function structures, which aim at increasing output corruption, we have proposed SK-Lock in [24]. SKG-Lock+ improves on this version, as will be detailed in this section. To the best of our knowledge, these solutions are the only ones to propose an improvement of a compound scheme, making good use of the advantages of both a point-function lock and key-gate insertion (cf. Figure 4d).

### 3.1. Framework

The structure of the SKG-Lock+ comprises two types of components, Switchable Key-Gates (SKGs) and a Switch Controller (SWC), as depicted in Figure 5. Two sets of key inputs are included in SKG-Lock+, the activation key ($K_A$) and the decoy key ($K_D$):

- $K_A$ is connected to the inserted SKGs:
    - Inserting the correct value of $K_A$ nullifies all SKGs and unlocks the circuit (this correct value is set by the designer during the design phase);
    - On the contrary, any wrong value of $K_A$ generates corruption through the SKGs (as long as they are triggered by the switch-signals);
    - The size of $K_A$ (denoted as $m$) indicates the effort of a brute-force attack.
- $K_D$ is connected to the switch controller:
    - Inserting the correct value of $K_D$ triggers the SKGs through the switch-signals;
    - On the contrary, any wrong value of $K_D$ disables some SKGs;
    - Note that the circuit can be unlocked irrespective of the value of $K_D$, inserting the correct value of $K_A$ is sufficient;

- – The size of $K_D$ (denoted as $n$) determines the security level against the SAT attack (as proven thereafter).

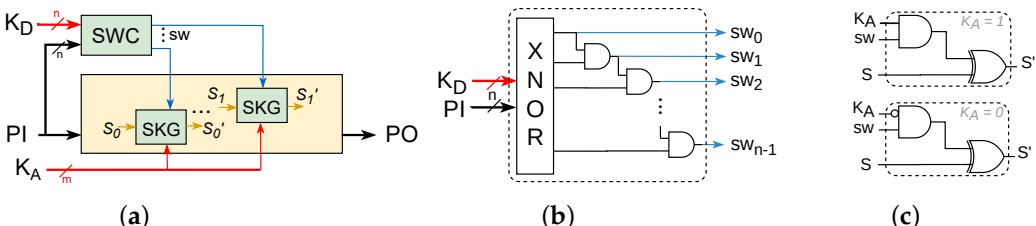

(a)  (b)  (c)

**Figure 5.** Original structure of SKG-Lock+ components: (**a**) general structure, (**b**) switch controller, (**c**) switchable key-gates.

$K_D$ and $K_A$ both come from a protected memory and are physically indistinguishable from the attacker's point of view. In other words, they are both controllable key inputs in the locked netlist used in oracle-guided attacks.

*3.2. Switchable Key-Gates*

3.2.1. Structure

The structure of the SKGs is depicted in Figure 5c. An SKG has three inputs: the signal $S$ from the locked circuit that is meant to be corrupted and two control signals—$K_A$ and the so-called switch-signal. Compared to a traditional XOR key-gate, beside the key input, an SKG has an additional control signal. Both control signals must be asserted in order to make an SKG corrupt $S$. Corruption indeed only happens when an incorrect $K_A$ value is inserted and '1' is set on the switch-signal (in case of SKGs with positive switch; note that SKGs with negative switch can be constructed with OR and XNOR gates). As a consequence, the choice of the switch-signal allows tuning the SKGs' ability to corrupt the circuit.

3.2.2. Insertion strategy $F_PLL$

Since SKG-Lock+ provides security against SAT-based attacks (cf. proof in Section 4.2), the criteria for SKG insertion strategy we chose is to maximize output corruption (One should notice that the key-gate insertion strategy is independent of the use of SKG-Lock+.

In other words, any existing insertion existing strategy can be used in the SKG-Lock+ framework.). The proposed key-gates insertion strategy based on fault analysis coming from probabilities computation $F_PLL$ consists of ranking signals in the circuit based on the co-called Output Corruption Score (OCS) to select signals for key-gate insertion.

The principle of the proposed strategy is to rank every signal of the circuit according to its OCS. For each signal, this metric reflects the impact on outputs if the signal is corrupted due to an inserted key-gate. Similar to FLL, this strategy emulates the corruption by inserting stuck-at-faults (s-a-f) on signals. Applying a wrong key bit is indeed equivalent to the activation of a stuck-at-0 (s-a-0) or stuck-at-1 (s-a-1) on the corresponding corrupted signal. Then, the impact of each given fault on outputs is measured by recording the difference in the outputs' probabilities (to be logic '1' (For the rest of the chapter, we use probability of a signal to refer to its probability to be logic '1'.)) with and without the fault as follows (cf. Figure 6). Calculating signals' probabilities in a netlist consists of propagating the probability of each signal from the circuit inputs (circuit primary inputs are assigned a probability of 0.5, as well as the outputs of FFs for sequential circuits) to the circuit outputs (cf. Figure 6a). A signal with an s-a-0 or s-a-1 changes its initial probability to 0 or 1, respectively, thereby influencing the probabilities of the signals in its fan-out (cf. Figure 6b,c).

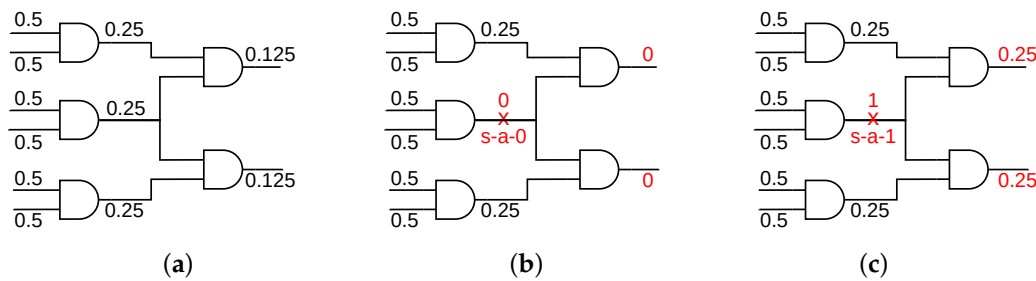

**Figure 6.** Signals probabilities when inserting stuck-at-faults: (**a**) initial probabilities, (**b**) modified probabilities if case of a s-a-0, (**c**) modified probabilities in case of a s-a-1.

The calculation of the OCS for each signal consists of computing the probability of outputs. Firstly, the probability of outputs in the original circuit is measured. Then, s-a-1/0 is inserted at that signal and the probability is recomputed. By comparing the probabilities before and after the fault insertion, one obtains the total probability difference $\Delta p_{saf}$ and the number of outputs that have their probability changed $n_{saf}$. $\Delta p_{saf}$ is the sum of absolute probability difference of each circuit output:

$$\Delta p_{saf} = \sum_{i=1}^{n_O} |p_i - p_{i_{saf}}| \tag{6}$$

where $n_O$ is the number of circuit outputs.

The OCS is calculated as:

$$OCS = \Delta p_{sa0} \times n_{sa0} + \Delta p_{sa1} \times n_{sa1} \tag{7}$$

Inserting key-gates at signals with high scores will impact most of the outputs for most of the input patterns, resulting in high output corruption coverage and corruption rate. One can notice that signals at circuit outputs always have an OCS of 1 since there is only one output affected and the total probability difference due to s-a-0 and s-a-1 $\Delta p_{sa0} + \Delta p_{sa1}$ is 1. On the other hand, internal signals with large fan-out potentially have a high OCS score and are favored by the strategy to achieve higher output corruption coverage.

Algorithm 1 describes the $F_PLL$ strategy, using key-gate insertion. The output corruption score of each signal is first calculated and signals are ranked according to their score in a descending order. After, signal selection starts from the one with the highest score. Here, an additional criterion is applied; the signals that have the same score as the previously chosen signal will not be selected for key-gate insertion. This is because signals that have the same score are structurally close to each other, such as signals connected by a buffer. Thus, selecting only one among these signals avoids series of key-gates.

The execution time of the $F_PLL$ strategy is essentially the signal ranking step. It can be estimated as:

$$T_{FPLL} = t_{prob} \times N \tag{8}$$

where $t_{prob}$ is the amount of time for calculating the output corruption score of a signal, $N$ is the number of signals. In comparison with FLL, which redoes the ranking each time a key-gate is inserted (cf. Equation (9)), $F_PLL$ only ranks signals once.

$$T_{FLL} = t_{sim} \times N \times K \tag{9}$$

Therefore, our strategy is more scalable than FLL. Furthermore, the time of the simulation needed by FLL may be quite long. If reconvergent paths are not taken into account, the probability-based computation of $F_PLL$ can be quite short (the version taking into account reconvergent paths takes obviously more time).

---

**Algorithm 1:** The $F_P LL$ strategy.

---
**Data:** netlist, keySize
**Result:** Locked netlist

1   signalList = [inputs, signals, outputs]
2   NbInsertedKeyGates = 0
3   **for** *I in signalList* **do**
4      Calculate *OCS* score of *I*
5   **end**
6   rankedSignalList = *rank*(signalList, descending order based on *OCS*)
7   **while** *NbInsertedKeyGates < keySize* **do**
8      Signal, Score = rankedSignalList.*pop*(0)
9      **if** *Score ≠ previousScore* **then**
10        Insert a key-gate at Signal
11        NbInsertedKeyGates += 1
12      **else**
13        continue (to avoid series of key-gates)
14      **end**
15   **end**
16   **return** locked netlist

---

### 3.3. Switch Controller

The SWC generates the switch-signals, thereby, determining the corruptibility of each SKG. Its inputs are an $n - bits$ key input $K_D$ and an equal number of circuit inputs, which can include primary inputs (PIs) and pseudo primary inputs (flip-flop outputs).

#### 3.3.1. Original SWC

A typical design for the SWC is depicted in Figure 5a. Its structure is a comparator, constructed with a row of XNOR gates and a cascade of AND gates. Its outputs are $n$ switch-signals, each of which from each signal in the AND cascade. The output at the end of the cascade $sw_{n-1}$ is essentially the output of a point-function between $K_D$ and circuit inputs. This switch-signal has a highly skewed probability, i.e., a low activity, since it switches $iff$ the value of the connected inputs is equal to $K_D$. Therefore, $sw_{n-1}$ presents low corruptibility but maximal complexity for the SAT attack. The SKG connected to it is referred to as the Lowest-Corruptibility SKG (LC-SKG). Other switch-signals have a higher probability (to be logic 1); thus, SKGs driven by them have higher corruptibility, compared to LC-SKG (the corruptibility $C$ of each SKG is the probability of its switch-signal: $C_{sw_0} = \frac{1}{2}$, $C_{sw_1} = \frac{1}{2^2}$, etc.). Consequently, the SKG with highest corruptibility is the one driven by $sw_0$. Depending on the designer's settings, several SKGs may be driven by the same switch-signal, and/or certain switch-signals may be unused (e.g., in cases when $n \neq m$). For example, to reduce output corruption, high-probability switch-signals may be left unused, and low-probability switch-signals can be connected to more SKGs. Experimental results will detail the possible trade-off between SAT resilience and output corruption according to the choice of switch-signals mapping.

#### 3.3.2. Improved SWC

However, with the original SWC structure presented in Figure 5a, there is an overlap in the input patterns that activate each switch-signal, resulting in a still quite large number of input patterns for which there is no corruption. A 3-bit example of the SWC structure used in SKG-Lock is illustrated in Figure 7a. In the corresponding switch-signals truth-tables, the rows present the primary inputs possibilities and the columns, the key values possibilities. With the original structure, due to overlap of input patterns that result in '1' for each switch-signal (cf. light orange boxes in the truth tables), there can only be half of all patterns (4 out of 8) that can lead to corruption. Therefore, it is necessary to improve

such an SWC structure in order to increase the number of patterns for which there can be corruption, as presented in Figure 7b. With this improved structure, different input patterns assert different switch-signals; hence, there are 7 out of 8 patterns for which there can be corruption. In cases when all inserted KA bits are incorrect, one can expect an output corruption rate up to almost 100%.

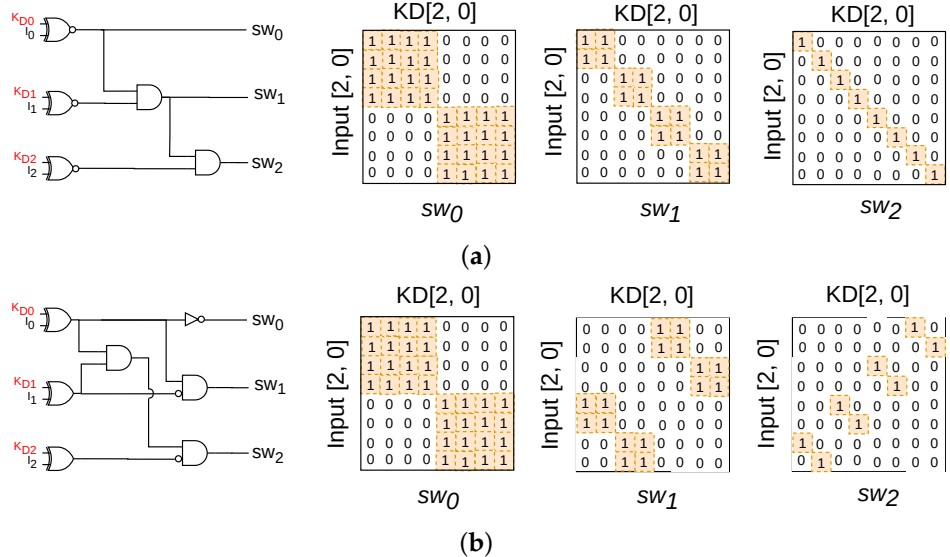

**Figure 7.** Switch controller structures (3-bits *KD* for example) and corresponding switch-signals truth tables: (**a**) the original structure, (**b**) the proposed improved structure avoiding overlapping in the inputs patterns that activate each switch-signal.

### 3.4. Locking Algorithm

Algorithm 2 describes how a netlist is locked with SKG-Lock+. The designer defines:

- The key sizes:
    - The size of $K_A$, hence the number of SKGs;
    - The size of $K_D$, hence the SAT-attack secure level;
- A list of signals for SKG insertion (note that if the netlist already contains key-gates, the designer can easily replace them with SKGs);
- A rule to map switch-signals to SKGs:
    - Which switch-signals to use: the SAT-attack-secure level (resp. output corruption) increases with increasing number of SKGs mapped to the switch-signal with the lowest (resp. highest) switching activity;
    - Which SKG to be mapped to which switch-signal, e.g., SKGs at strategic locations could be mapped with high-activity switch-signals to increase output corruption.

A locked netlist is then formed from the original netlist, by inserting SKGs and the SWC block, mapping circuit inputs to the SWC and switch-signals to SKGs, as depicted on a toy example in Figure 8. Note that this insertion process is, although linear to the key size, almost instantaneous and independent of the size of the circuit. The total procedure computation time also includes the step of creating the signals list for the insertion of the SKGs, which is more time consuming than the insertion itself.

---

**Algorithm 2:** SKG-Lock+ locking algorithm.

**Data:** Netlist, size of $K_A$ (m), size of $K_D$ (n), candidate signals list, SKG-*sw* mapping rule

**Result:** Locked netlist

1    *Locked netlist = Netlist*;

2    Add $n − bit$ $K_D$ and $m − bit$ $K_A$ to *Locked Netlist* primary inputs;

3    Insert SWC block in *Locked Netlist* with 2*n* inputs and *n* outputs;

4    Map *n* SWC inputs to $K_D$;

5    Map other *n* SWC inputs to *n* randomly chosen primary inputs;

6    **for** *each* bit $K_{Ai}$ **in** $K_A$ **do**

7      |   Insert an SKG at $signal_i$ of candidate signals list in *Locked Netlist*;

8      |   Map $k_a$ control input signal to $K_{Ai}$;

9      |   Map *switch* control input signal to one *sw* according to SKG-*sw* mapping rule;

10   **end**

11   Return *Locked netlist*;

---

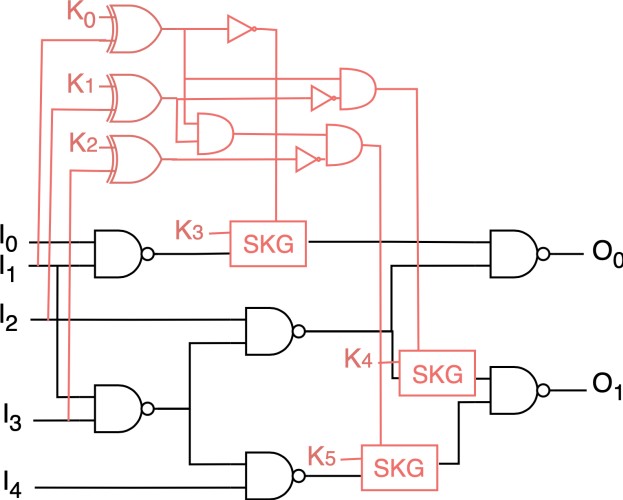

**Figure 8.** SKG-Lock+ insertion on a toy example.

### 3.5. Area Overhead

The area overhead of the SKG-Lock+ is dependant of the key size. The number of SKGs is indeed equal to the size of $K_A$ (each SKG generates 1 XOR/XNOR gate and 1 AND gate), and the size of the SWC is proportional to the size of $K_D$. Compared to the original structure, the proposed improved SWC structure requires a little more area; for an n-bit SWC, it uses additional n-2 AND gates and its number of gates is estimated as $n$ XOR/XNOR gates and $2 \times n - 2$ AND gates. Additional XOR-gates are also present in case of an obfuscated SWC.

By comparison with methods of the same type, for the same SAT resilience, SKG-Lock+ has a comparable cost in area to CAS-Lock ($2 \times n + 1$ XOR/XNOR gates and $2 \times n + 1$ AND/OR gates), but much less than SFLL-HD, which contains HD counters.

## 4. Security Analysis of SKG-Lock+ Against Oracle Guided Attacks

### 4.1. Key Sensitization Attack

The key sensitization attack [8] is able to counter key-gate-based logic locking by targeting each key-gate and sensitizing individual key-input. With SKG-Lock+, for each SKG, there is a convergence path between its $K_A$ signal and its switch-signal. Furthermore, switch-signals are convergent with (one to) several $K_D$ signals. Since $K_D$ key-inputs and $K_A$ key-inputs are indistinguishable from the attacker's point of view, there is interference among key-inputs. In addition, an interference-based strategy [10] can be used as the

SKG insertion strategy to further provide interference among $K_A$ key-inputs. Therefore, SKG-Lock+ is resistant against the key sensitization attack.

### 4.2. SAT Attack

Without loss of generality, we can assume that the size of $K_D$ and circuit PI is $n$; the correct value of each bit of $K_A$ is 1 for every SKG; and each SKG is inserted at a different circuit output so that any corruption is observed at an output. A DIP produced at the $i$-th iteration by the SAT attack is denoted as $X_i$. Let us denote $N$ as the number of iterations of an SAT attack.

Let us consider the case of SKG-Lock, including a switch controller and only one SKG controlled by $sw_{n-1}$. Wrong key values that can be ruled out by a DIP $X_i$ satisfying the following condition:

$$(K_A = 0) \wedge (\vec{K_D} = \vec{X_i}) \tag{10}$$

For any given $X_i$, there is one way to select $K_A$ and one way to select $K_D$ to satisfy the condition in Equation (10). Thus, each iteration identifies only one wrong key value. Hence, the number of iterations required by the SAT attack to eliminate all ($2^n$) wrong key values is $N = 2^n$. The circuit is $n$-secure against an SAT attack.

Following the same assumptions, let us now consider the case where the circuit is locked with two SKGs: one SKG is driven by $sw_{n-1}$ and another SKG is driven by $sw_{n-2}$. The condition for any wrong key value to be identified by a given $X_i$ is:

$$\left[ (\vec{K_A}[0] = 0, \vec{K_A}[1] \in \mathbb{B}) \wedge (\vec{K_D} = \vec{X_i}) \right] \vee$$
$$\left[ (\vec{K_A}[0:1] = \vec{10}) \wedge (\vec{K_D}[0:n-2] = \vec{X_i}[0:n-2]) \right] \tag{11}$$

Thus, when $\vec{K_A}[0] = 0$, the set of wrong key eliminated by $X_i$ has the following form:

$$(\vec{K_A}[0] = 0, \vec{K_A}[1] \in \mathbb{B}, \vec{K_D} = \vec{X_i}) \tag{12}$$

There is a one-to-one matching between $K_D$ and $X_i$. Thus, any input pattern can be selected as a DIP to identify a unique set of wrong keys in the form of (12). Therefore, the total number of SAT iterations is $N = 2^n$. The circuit is $n$-secure against an SAT attack.

Figure 9 illustrates this proof with two examples of truth tables. In these examples, there is a 2-bit $K_A$, connected to two SKGs, and a 3-bit $K_D$, connected to a 3-bit SWC. Each truth table is divided into four sub-parts according to the four possible values of $K_A$ (and in each sub-part, eight possible values for $K_D$). The two control signals of the first SKG are $K_A[0]$ and $sw_{n-1} = sw_2$. The two control signals of the second SKG are $K_A[1]$ and $sw_1$ in Figure 9a or $sw_0$ in Figure 9b. In both figures, the point-function behavior appears when $K_A[0]$ is not correct but $K_A[1]$ is correct (third sub-part of the truth tables), which imposes on the SAT attack at least $2^n (n = 3)$ iterations to eliminate all wrong key values in this sub-part. Furthermore, when $K_A[1]$ is not correct (first and second sub-part), the corruption (of the signals on which the SKGs are inserted) is increased, depending on which switch signal is used.

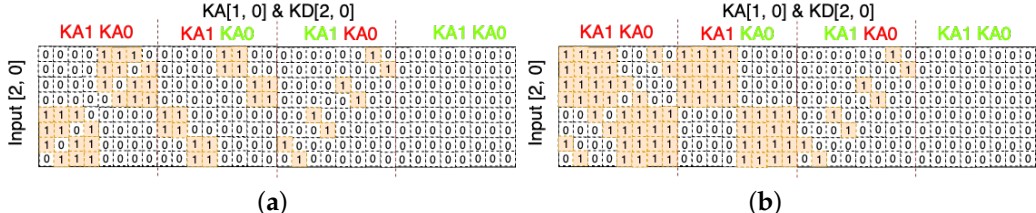

**Figure 9.** The truth tables representing corruption (1 in light orange) with the proposed improved SWC structure, depending on the possible key values (correct bits in green and incorrect bits in red): (**a**) one SKG connected to $sw_2$ and one SKG connected to $sw_1$, (**b**) one SKG connected to $sw_2$ and one SKG connected to $sw_0$.

More generally, the presented proof holds for the case where there are more than two SKGs. $n$-secure SAT-resilience level is achieved as long as there is at least one SKG connected with $sw_{n-1}$. The achieved SAT resilience depends on the size of the SWC.

### 4.3. AppSAT Attack

The AppSAT attack [15] aims to find an low-corruption key, i.e., a key value for which the output corruption of the locked circuit is very low. However, AppSAT is not effective against logic locking techniques when different key values correspond to different amounts of output corruption [52]. This is the case for SKG-Lock+ since each key value indicates a different set of incorrect key bits for SKGs and each SKG has a different corruptibility. The key value that results in the lowest corruption is the one in which only the key bit of the lowest corruptibility SKG is incorrect. Therefore, one can expect that AppSAT on SKG-Lock+ returns a key value that has several wrong key bits, which leads to considerable corruption.

### 4.4. Bypass Attack

The Bypass attack [39] aims to construct a bypass circuit to correct the corrupted outputs of a locked circuit. The attack builds a miter circuit (with two copies of the locked circuit applied with two random key values) to find all input patterns that cause corrupted outputs. In SKG-Lock+, two random key values may contain same wrong key bits for SKGs. Hence, the two locked copies may have the same wrong outputs for several input patterns, which then would go unnoticed by the attack. Furthermore, an incorrect key value could lead to a significant output corruption rate and coverage, which results in an impractically large bypass circuit. Therefore, the attack is not efficient against SKG-Lock+.

## 5. Experimental Results

To evaluate $F_P LL$ and SKG-Lock+, we implemented them on ISCAS'85, ISCAS'89, MCNC or/and ITC'99 benchmarks [53,54]. The experiments were executed on an 8-core Intel processor running at 1.90 GHz with 16 GB RAM.

### 5.1. $F_P LL$

For measuring probabilities, we used the Signal Probability Reliability Analysis (SPRA) tool [55]. In our experiments, we chose not to use the option that takes into account reconvergent paths. Without this option, the program takes much less time, despite slightly less accurate measurement, than with this option [56]. Nevertheless, as will be shown, optimal results for output corruption were nevertheless obtained, showing that this option is not useful.

To evaluate the output corruption of $F_P LL$, we implemented XOR/XNOR key-gate insertion with our strategy, FLL, and RLL, each on six benchmarks. For each benchmark, the number of inserted key-gates is 5% of the number of gates in the circuit (130 for i8, 124 for c5315, 178 for seq, 186 for c7552, 269 for apex4, 336 for des). For this evaluation, each circuit was simulated with 100 wrong key values, each with 1000 random input patterns. The results are presented in Figure 10. As $F_P LL$ and FLL are optimized for output corruption, both achieve optimal results in all metrics. For $F_P LL$, most circuits achieved output corruptibility from 40% to the optimum 50%. Its output corruptibility is equivalent with that of FLL; the results are slightly better for four circuits. It also has maximum output corruption coverage due to the fact that it favors signals that effect the most outputs as possible; the results are better than that of FLL for two circuits. It achieves 100% corruption rate in all circuits, which is equal to FLL and better than RLL for one circuit. $F_P LL$ performs significantly better than RLL, especially in output corruptibility and corruption coverage.

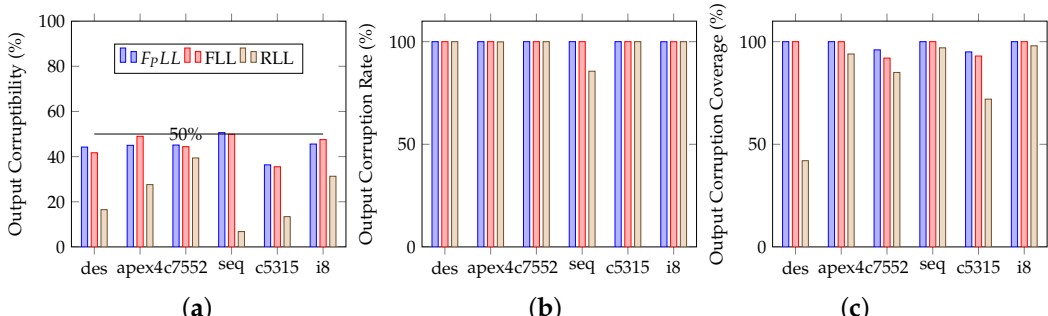

**Figure 10.** Output corruption evaluation of $F_PLL$ and comparison with FLL and RLL (benchmarks are in decreasing size order): (**a**) uutput corruptibility, (**b**) output corruption rate, (**c**) output corruption coverage.

Table 1 shows the execution time results in an increasing order. In general, larger circuits, i.e., circuits with a higher number of signals (cf. second column), require more time than smaller ones; however, it is a non-linear relation; the runtime scales up faster than the size of the circuit. This is because the probability measurement runtime depends on the circuit size. Nevertheless, for small circuits, the $F_PLL$ strategy finished in a matter of minutes.

A preliminary comparison with FLL shows that, regarding benchmark c7552, $F_PLL$ finished in 10 min whereas it is reported in [7] that FLL took two hours, which makes $F_PLL$ 92% faster. Nevertheless, the comparison may not be relevant due to possible discrepancies in time measurements, such as the workstation specifications. To better compare $F_PLL$ and FLL, we made experiments on three small benchmarks (c432, c1355 and i9), which showed that $F_PLL$ was 85% to 95% faster, confirming previous data.

In summmary, $F_PLL$ is as efficient as FLL in terms of output corruption, with significantly shorter execution time.

**Table 1.** $F_PLL$ runtime.

| Bench | Nb Signals | Runtime (s) |
|---|---|---|
| c5315 | 2485 | 225.35 |
| i8 | 2597 | 205.15 |
| s5378 | 3050 | 495.23 |
| seq | 3560 | 469.87 |
| c7552 | 3720 | 605.5 |
| apex4 | 5370 | 1609.4 |
| des | 6729 | 3340.83 |
| s9234 | 5844 | 3729.66 |
| s13207 | 8729 | 12,351.7 |
| b15_C | 8922 | 14,337.47 |
| b14_C | 10,098 | 19,663.67 |
| s15850 | 10,397 | 25,355.29 |

### *5.2. SKG-Lock+*

We set in each benchmark an equal size of $K_A$ and $K_D$, $n = m$, the total key size is therefore $2n$. The $n$ inputs of the SWC were randomly selected. $n$ SKGs were inserted and $n$ switch-signals were used, each of which was driving each SKG.

#### 5.2.1. Security Evaluation

As mentioned in Section 4.2, the security level against the SAT attack depends directly on the size of $K_D$ (i.e., the number of inputs connected to the switch controller). Therefore, the expected number of iterations for each benchmark is $2^n$ in order to be $n$-secure against the SAT attack.

The evaluation of SAT resilience of the base configuration of SKG-Lock+ with increasing key size is shown in Figure 11 (for the sake of understanding the trends, we made experiments with small key sizes). For all benchmarks, the number of SAT iterations is bigger than the expected number for *n*-secure. The cause of the extra iterations (could be thousands of iterations) stems from the locations of inserted SKGs. Propagating SKGs corruption to circuit outputs indeed involves controlling several inputs. Thus, inputs that are not connected to the switch controller may also be taken into account (in addition to the connected ones) when the attack identifies DIPs.

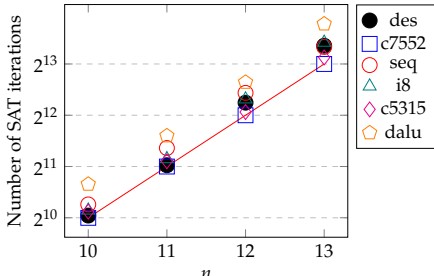

**Figure 11.** Evaluation of SAT resilience vs. key size.

The level of security against approximate attacks depends on the key size (as for the SAT attack). Furthermore, the accuracy of the approximate key potentially found by the attack—directly related to the output corruption generated by this key—is also of interest. We applied AppSAT on SKG-Lock+ 32-secure against SAT (i.e., $n = 32$). We used the same attack configuration as in [25]: 50 random queries were applied after every 12 SAT iterations and the settlement threshold was 5 and ran the attack ten times on each benchmark.

The results in Table 2 show the output corruption of the circuit when applied with the key values returned from AppSAT. We observed that the $K_A$ part of the AppSAT keys contains several wrong key bits. To measure the output corruption with the produced keys, for each key value, we applied 1,000,000 random input patterns and compared the outputs observed from the locked circuit to the golden outputs. It is apparent that AppSAT failed to reduce the output corruption rate of SKG-Lock to a point-function corruptibility ($1/2^{32}$ in this case). The observed corruption rates range from 0.1% to 6%, indicating that the circuit, if run at 1MHz, may produce several thousands of errors each second. Moreover, we also observed sufficient output corruption coverage on several benchmarks.

**Table 2.** AppSAT attack result on SKG-Lock+ ($n = 32$).

| Bench | Output Corruption Due to AppSAT Key | |
|---|---|---|
| | **Output Corruption Rate (%)** | **Output Corruption Coverage (%)** |
| des | 0.14 | 24.49 |
| c7552 | 3.1 | 25.23 |
| i8 | 1.03 | 98.76 |
| c5315 | 6.64 | 32.52 |
| dalu | 2.18 | 100 |

Note that we have not conducted experiments on some recent proposed oracle-guided attacks, dedicated to "classical" compound schemes or CAC schemes, SKG-Lock+ being in nature a structurally different approach (note also that not all attack frameworks are made available by the authors). Attacks dedicated to CAC schemes indeed often exploit the protected input pattern(s) of those schemes, feature that SKG-Lock+ does not have. Furthermore, output corruption variation is far more linear with SKGLock+ than with classical compound schemes, in which the variation in output corruption is the feature exploited [42]. Making further experiment to validate the efficiency of SKG-Lock+ against these attacks is nevertheless part of our future work.

### 5.2.2. Output Corruption

We implemented SKG-Lock+ ($n = 64$) along with $F_PLL$ and provide a comparison with previously proposed SKG-Lock (combined with FLL), for the same SAT resilience level, and with one of the most recently improved PSLL schemes, CAS-Lock. The CAS-Lock block contains a cascade of AND gates followed by an OR gate, which allows the highest corruptibility possible among all configurations of CAS-Lock, and is inserted at a high-controllability signal in the circuit. For SKG-Lock, the mapping between the SKGs and the switch-signals was conducted randomly since the input was a locked circuit with FLL with no further information about the fault-impact of each signal used to perform the locking process. In other words, all switch-signals were used, and the choice of which switch signal to connect to and which SKG was random. Conversely, thanks to the use of the $F_PLL$ insertion strategy—and therefore the knowledge of all signals' OCS—along with SKG-Lock+, it was possible to choose a more pertinent SKG-sw mapping rule. To maximize corruption, we chose to map switch-signals with decreasing corruptibility to signals with decreasing OCS.

Table 3 presents the results for output corruption rate and output corruption coverage. As can be seen, SKG-Lock+ achieves far better results than CAS-Lock in both metrics, and slightly better results that SKG-Lock for the output corruption rate. One can observe that, due to the scattering of SKGs throughout the circuit, SKG-Lock and SKG-Lock+ are able to affect all circuit outputs in several cases. Conversely, CAS-Lock only corrupts one signal in the circuit, thereby affecting only a few outputs. Furthermore, since $F_PLL$ and FLL have been shown to produce somewhat equivalent output corruption, one can therefore deduce that the better results of SKG-Lock stem for the new SWC structure. In terms of output corruptibility, SKG-Lock+ with $F_PLL$ obtains significantly better results that SKG-Lock with FLL and CAS-Lock. For example, for the *des* benchmark, SKG-Lock+ produces a 16.5% output corruptibility, whereas SKG-Lock produces 1% and CAS-Lock 0.1%. It should be noted that, in comparison, for the same SAT resilience, SARLock, Anti-SAT, and SFLL-HD$^0$ have a corruption rate of $1/2^{64} = 5.4e^{-18}\%$.

**Table 3.** Output corruption evaluation on SKG-Lock+ ($n = 64$).

| Bench | Output Corruption Rate (%) | | | Output Corruption Coverage (%) | | |
|---|---|---|---|---|---|---|
| | SKG-Lock+ $F_PLL$ | SKG-Lock FLL | CAS-Lock | SKG-Lock+ $F_PLL$ | SKG-Lock FLL | CAS-Lock |
| des | 48.5 | 49.4 | 23.47 | 100 | 100 | 0.8 |
| c7552 | 50 | 49.6 | 8.79 | 61.68 | 58.88 | 0.93 |
| c5315 | 49.5 | 23.9 | 12.46 | 63.42 | 78.05 | 1.63 |
| i8 | 36 | 11.6 | 8.89 | 100 | 100 | 1.235 |
| dalu | 12.24 | 31 | 3.12 | 100 | 100 | 25 |
| Average | 39.25 | 33.1 | 11.35 | 85.02 | 87.39 | 5.59 |

We also measured the maximum output corruption that SKG-Lock+ can produce. It is the case when all inserted $K_A$ bits are wrong. The results, reported in Table 4, show that in several cases, the output corruption rate reaches close to 100%. The high output corruption rate is achieved thanks to the proposed improved SWC structure.

**Table 4.** Maximum output corruption of SKG-Lock+ ($n = 64$).

| Bench | Output Corruption Rate (%) | | Output Corruption Coverage (%) | |
|---|---|---|---|---|
| | SKG-Lock+ $F_PLL$ | SKG-Lock FLL | SKG-Lock+ $F_PLL$ | SKG-Lock FLL |
| des | 97 | 98.61 | 100 | 100 |
| c7552 | 99.9 | 99.9 | 64.5 | 86.9 |
| c5315 | 99 | 47.87 | 68.3 | 78 |
| i8 | 71.63 | 22.16 | 100 | 100 |
| dalu | 24.35 | 61.92 | 100 | 100 |

We further investigated the relation between SAT resilience and output corruption. To create even higher SAT-resilient configurations of SKG-Lock+, we restricted the number of switch-signals and the selection was made according to the decreasing corruptibility order. Thus, the number of switch-signals can range from $n$ down to 1, where $n$ stands for the base configuration (all switch-signals are used) and 1 stands for the lowest output corruption (only $sw_{n-1}$ is used for all SKGs).

These results are reported in Figure 12. Once again, the number of SAT iterations is higher than expected. Furthermore, also as expected, lower output-corruption configurations are inclined to have higher gain in iterations. Nevertheless, using multiple times $sw_{n-1}$ does not necessarily lead to a significant increase in the number of iterations (between $2n$ and $2n + 1$ for most benchmarks), but significantly reduces output corruption rate. Note that output corruption coverage is not impacted since it depends on the locations of the SKGs rather than the mapping between SKGs and the SWC. These results confirm that mapping each switch-signal to a different SKG (cf. first column in Figure 12) is the most optimal way to achieve both high SAT resilience and high output corruption.

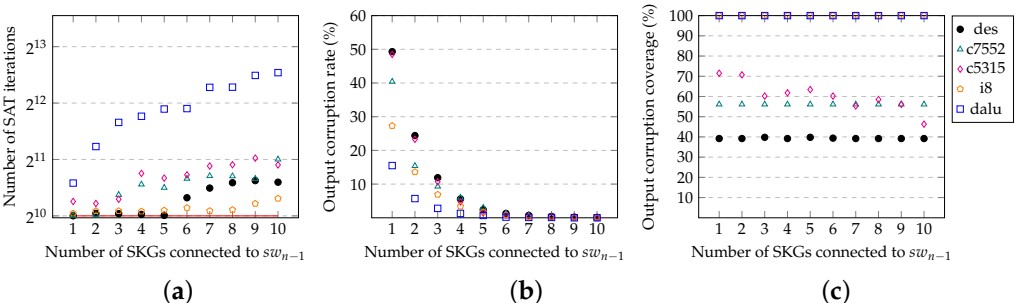

**Figure 12.** Evaluation of SAT resilience and output corruption according the switch-signals used: (**a**) Number of SAT iterations, (**b**) Output corruption rate, (**c**) Output corruption coverage.

### 5.2.3. Overhead Evaluation

The overhead of SKG-Lock+ was evaluated in terms of area, power, and delay overheads, on the synthesized netlist using a STMicroelectronics 65 nm CMOS process, using Design Compiler. The benchmarks were implemented with SKG-Lock+ with $F_PLL$ insertion strategy (and no re-synthesis step).

Figure 13a shows the overhead on different benchmarks as a 32-bit key. For medium benchmarks such as des and c7552, the average overhead is less than 10%. For smaller benchmarks, the overhead is bigger. Figure 13b presents the overhead as the key size increases on benchmark c7552. The area and power overhead scale linearly with the key size. The delay overhead is equivalent to the number of SKGs inserted at the critical path of the circuit. Thus, it does not depend directly on the key size. Note that, to limit the delay overhead, an additional constraint to prevent the insertion of key-gates in critical paths can easily be incorporated in the strategy.

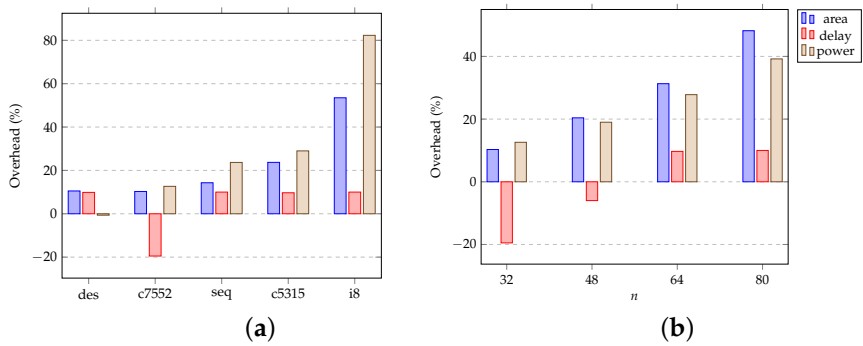

**Figure 13.** Overhead evaluation of SKG-Lock+: (**a**) overhead with $n = 32$ (benchmarks are in decreasing size order), (**b**) overhead evaluation vs. key size (benchmark is c7552).

### 5.2.4. Comparison with Related Works

The comparison of SAT resilience level of SKG-Lock+ with related works is presented in Table 5. For a fair comparison, the logic locking structure is connected with $n$ circuit inputs, which corresponds to a key size of $n$ bits in SFLL, and $2n$ bits CAS-Lock, SKG-Lock, and SKG-Lock+ (if $m = n$) and all possible values above $n$ for the compound scheme (we can assume $2n$ for sufficient output corruption). Whereas initial PSLL techniques sacrificed output corruption for SAT resilience, CAC techniques such as SFLL trade resilience for a little more corruption. As for CAS-Lock, SKG-Lock, and SKG-Lock+, they achieve $n$-secure without compromising output corruption.

Comparison with other types of methods, e.g., based on scan-chain protection [32,33], is also of interest and part of our future work.

**Table 5.** Comparison of SKG-Lock+ with related works.

| Techniques | Compound | CAC | | | Improved PSLL | | |
|---|---|---|---|---|---|---|---|
| | | **SFLL-HD** | **SFLL-flex** | **CAS-Lock** | **SKG-Lock** | **SKG-Lock+** | |
| SAT resilience level | $n$ | $n - \lceil log_2 \binom{n}{h} \rceil$ | $n - \lceil log_2 c \rceil$ | $n$ | $\geq n$ [†] | $\geq n$ [†] | |
| Output corruption | High | Low | Low | Medium | High | High(er) | |
| Average overhead | Low | High | High | Low | Low | Low | |

[†] Higher level can be achieved with lower-corruption configurations.

### 6. Discussion and Future Work

*What about manufacturing test?* Back to the threat model, with logic locking, outsourced stages (including fabricating, testing, and packaging) are completed on locked ICs, by untrustworthy offshore foundry and OSAT. The foundry fabricates silicon ICs of the locked design. The manufacturing test is performed before IC activation/on locked ICs also. This is possible since the manufacturing test, which is essentially a structural test, can be performed irrespective of circuit functionality [57]. Furthermore, test pattern generation can be conducted without constraints on the key-bits, so the faults in the SWC of SKG-Lock+ and those on the key lines of the SKGs can be tested without any problem, the controllability of the signals in the logic cones driven by an SKG are also improved. In summary, the fault coverage is not decreased by the addition of SKG-Lock+, it can even be increased in some cases.

*What about hardware Trojan horses?* Considering that in the threat model considered, the foundry is the primary attacker a designer wants to protect against, it is reasonable to assume that they would also test the manufactured ICs against potential inserted hardware Trojan horses, especially those inserted especially for leaking the secret key [58].

*What about oracle-less attacks?* As already explained, oracle-less attacks include removal attack and synthesis-based attacks.

Removal attacks consist of analyzing the circuit structure to detect and remove the protection. The SWC is a critical component of SKG-Lock+; if removed, all SKGs could be switched off. Structural analysis methods such as skewed-probability signal identification, circuit partitioning, and fanin analysis have been used to identify similar SAT-resistant blocks. SKG-Lock+ generates structural entanglement between the SWC and the locked circuit thanks to its multiple connections with the SKGs, in contrast to previous point-function-based techniques. Thus, a partitioning algorithm [43] cannot be used to separate this block from the circuit. Fanin analysis [44] is used to find the output of an obfuscated point-function block due to the convergence of all key-inputs. In SKG-Lock+, while $K_D$ key-inputs are connected to the SWC, $K_A$ key-inputs are connected to SKGs, which are inserted in the locked circuit. Therefore, finding the convergence points of all key-inputs leads an attacker to signals in the locked circuit rather than in the SWC. Since the SWC is based on a point function, it contains a few signals with highly skewed probabilities. XOR

key-gates could be added to balance the probabilities of signals (without compromising SAT resilience).

Furthermore, one should notice that a subsequent re-synthesis step may transform the recognizable structures of both the SWC and the SKGs, and merge them with neighbor gates in the locked circuit, thereby preventing an attacker from detecting them to remove them. To corroborate this claim, we performed the following experiment. We have re-synthesized the same locked benchmark with SKG-Lock+ (on benchmarks c2670) multiple times, using different optimization parameters and delay constraints. We obtained 521 different re-synthesized netlists, for which we observed the types of the gates connected to $K_A$ and $K_D$ bits. These preliminary results are summarized in Table 6. As one can see, $K_A$ key bits, initially only connected to AND gates or inverters, are now also connected to OR gates mainly. Even more interesting, $K_D$ key bits, initially connected to XNOR gates, are now connected mainly to inverters, AND and OR gates. These preliminary results confirm that a re-synthesis step should prevent an attacker from distinguishing the SWC and $K_A$ from $K_D$ bits.

More experiments are nevertheless needed, especially since re-synthesis has recently been attacked, to some extent, by SAIL [59] (also combined with functional analysis [60]), SnapShot [61], and OMLA [62], which rely on machine learning model to retrieve the—small, localized and predictable—structural changes induced by re-synthesis. These attacks are nevertheless currently only dedicated to key-gate-based logic locking methods.

**Table 6.** Evaluation of SKG-Lock+ against removal attacks after re-synthesis ($n = 64$).

| Gate Type | $K_A$ | $K_D$ |
|:---:|:---:|:---:|
| NOT | 3879 | 6477 |
| BUF | 4 | 0 |
| NAND | 0 | 0 |
| NOR | 0 | 0 |
| AND | 5950 | 17,536 |
| OR | 7678 | 14,288 |
| XOR | 0 | 94 |
| XNOR | 0 | 94 |

SKGLock+ should also be secure against the FALL and CLIC-A attacks [16,51] since it does not rely on a protected input pattern, which is the feature, common to TTLock and SFLL among others, that is exploited by these attacks.

Regarding synthesis-based attacks, the SKGs structure is not secure against the SCOPE attack [49] (cf. Figure 14a). However, thanks to a simple countermeasure, the attack can be thwarted. As shown in Figure 14b, thanks to an additional XOR/XNOR gate (making the SKGs to be controlled by two key-inputs, one for a $K_A$ bit and the other for a $K_D$ bit) assigning either value to a key-input only removes the additional XOR/XNOR gate. Therefore, it is challenging for the attack to determine which circuit copy is more optimized, hence, making it unable to recover the key bit.

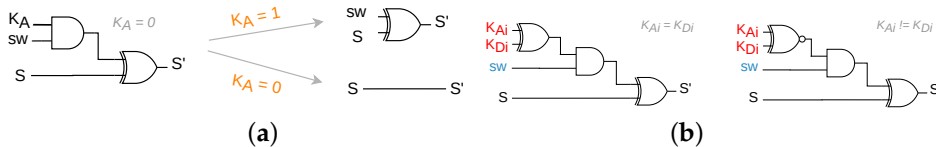

(a)　　　　　　　　　　　　　　　(b)

**Figure 14.** Vulnerability and countermeasure of SKG-Lock+ against the SCOPE attack: (**a**) vulnerability of SKG-Lock, (**b**) countermeasure based on a new SKG structure.

In summary, SKG-Lock+ is, to some extent, secure against most oracle-less attacks. Part of our future work is to better assess SKG-Lock+ against these attacks, with also the help of the Valkyrie assessment tool [63] to better assess potential structural vulnerabilities.

*What about RTL/FSM locking?* FSM/RTL locking was introduced around the same time as gate-level logic locking [64] and has also been a very prolific subject of study to this date [65,66]. Among other things, it was also attacked by SAT-based attacks. A thorough comparison of SKG-Lock+ with such proposals could be of interest. Interested readers about FSM/RTL locking can refer to [67].

## 7. Conclusions

In this paper, we proposed an improvement of a previously proposed logic locking technique, referred to as SKG-Lock+, for preventing locked circuit usage without requiring an unlock procedure from the designer. SKG-Lock and SKG-Lock+ based on novel switchable key-gates and a switch controller, controlled by decoy key-inputs. Futher, an improved structure of the switch-controller combined with a newly proposed key-gates insertion strategy $F_P LL$ make SKG-Lock+ superior to its preliminary version in terms of the output corruption reached.Futhermore, SKG-Lock+ is highly configurable: one can choose the exact number of key-bits dedicated to SAT protection—hence, the SAT resilience level—and those to output corruption, as well as the key-gates insertion strategy (SKG-Lock+ is easily adaptable to any strategy), and the mapping rule between its two mains components to further tune output corruption at will.

To the best of our knowledge, SKG-Lock and SKG-Lock+ are the first methods of their kind being enhanced compound structures that take advantage of the combined benefits of both a point-function and the insertion of key-gates (hence, corruption at multiple points of insertion) with intrinsic entanglement between the two.

**Author Contributions:** Q.-L.N. proposed the ideas and implemented the framework including obtaining the experimental results. S.D., M.-L.F and B.R. gave technical feedbacks. S.D. and Q.-L.N. wrote the manuscript. M.-L.F. and B.R. reviewed the manuscript and discussed the writing. All authors have read and agreed to the published version of the manuscript.

**Funding:** This research was funded by project MOOSIC ANR-18-CE39-0005 of the French National Research Agency (ANR).

**Institutional Review Board Statement:** Not applicable.

**Informed Consent Statement:** Not applicable.

**Data Availability Statement:** Not applicable.

**Conflicts of Interest:** The authors declare no conflict of interest. The funders had no role in the design of the study; in the collection, analyses, or interpretation of data; in the writing of the manuscript; or in the decision to publish the results.

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
