# Peer review of "SKG-Lock+: A Provably Secure Logic Locking SchemeCreating Significant Output Corruption"

_electronics, doi:10.3390/electronics11233906_

Round 1

Reviewer 1 Report

This manuscript presents a new logic locking technique called SKG-Lock+, which is an improved version of a previously proposed technique by the authors. The method is well considered and explained by citing and discussing a number of attacks. The reviewer thinks there are no apparent flaws in the proposed method.

Although the manuscript can be acceptable in the present form, the reviewer has small comments on the presentation of Fig. 9.

1. If possible, all of the names of benchmarks should be listed up in the x-axis.

2. Output corruptibility of 50% should be clearly described in Fig. 9 (a), as it is the target value.

3. Bar charts should not start at non-zero value (Fig. 9 (b) and (c)) i.e., the length of a bar should be proportional to the corresponding value.

Author Response

Dear reviewer,

Thank you for the comment, figure 9 has been modified according to your expectations:

1. All of the names of benchmarks are listed up in the x-axis.

2. An axis has been added in Fig. 9 (a) to better emphasize the target value of 50%.

3. Bar charts all start at zero.

Reviewer 2 Report

In this paper, the authors describe a hardware-locking scheme to prevent cloning. The two contributions are well supported and discussed.

The paper is very interesting and even though it carries the required complexity it is very useful and well-written.

Minor remarks:

@139, which need to me=>which need to be

@614 not only not decreased = > not only increased

Also, I would suggest the authors add an MWE (Minimal Working example), that illustrates how the technique works. For example, use a demo circuit with 5 levels of logic hierarchy and 10 gates. Finally, when referring to time 'far shorter computation time'  you should give all necessary details of how the time measurements were done, if the execution workstation had the same specifications and if the implementation was performed using the same programming language/the same number of threads, and so on.

Author Response

Dear reviewer, 

Thanks for your feed back.

1)The first typo has obviously been corrected according to your comment. Regarding the second typo, changing "not decreased" into "increased" would, in our opinion, change the meaning of what we meant to say. But since "not only not" was poorly phrased and could lead to confusion, we chose to change the phrase as follows:

In summary, the fault coverage is not decreased by the addition of SKG-Lock+, it can even be increased in some cases.

2) A figure has been added on page 12 (new Figure 8)

3) Thank you for pointing out this flaw. We chose to make further experiments and rephrase this part as follows.

A preliminary comparison with FLL shows that, regarding benchmark c7552, FPLL finished in 10 minutes whereas it is reported in [7] that FLL took two hours, which makes FPLL 92% faster. Nevertheless, the comparison may not be relevant due to possible discrepancies in time measurements, such as the workstation specifications. To better compare FPLL and FLL, we made experiments on three small benchmarks (c432, c1355 and i9), which showed that FPLL was 85% to 95% faster, confirming previous data.

We hope that these changes suit you.